mental health; mental health prevention; mental health promotion strategies; young people; sub-Saharan Africa

**Corresponding author:**
Tholene Sodi;
Email: tholene.sodi@ul.ac.za

# Types and effectiveness of mental health promotion programmes for young people in sub-Saharan Africa: A systematic review

Tholene Sodi[1] , Katlego Rantho[1], Frans Koketso Matlakala[2], Pamela Wadende[3] , Deborah Ikhile[4], Samuel Oluwasogo Dada[5], Diana Frost[5], Paulette Henry[6], Utek Ishaku[7], Michael Obeng Brown[5], David Musoke[8], Shai Nkoana[1], Talamo Phochana[1], Dung Jidong[9], Juliet Pwajok[7] , Toluwalope Awokoya[5], Elma Rejoice Banyen[5] and Linda Gibson[5]

[1]Department of Psychology, University of Limpopo, Sovenga, South Africa; [2]Research and Innovation Directorate, University of Venda, Thohoyandou, South Africa; [3]Department of Educational Psychology, Early Childhood and Special Needs Education, Kisii University, Kisii, Kenya; [4]Leicester Diabetes Centre, University of Leicester, Leicester, UK; [5]Institute of Health and Allied Professions, Nottingham Trent University, Nottingham, UK; [6]Faculty of Social Sciences, University of Guyana, Georgetown, Guyana; [7]Department of Psychology, University of Jos, Jos, Nigeria; [8]Department of Disease Control and Environmental Health, Makerere University, Kampala, Uganda and [9]Department of Psychology, University of Manchester, Manchester, UK

## Abstract

Studies show that mental health promotion is an effective strategy that can reduce the burden of mental health disorders and improve overall well-being in both children and adults. In addition to promoting high levels of mental well-being and preventing the onset of mental illness, these mental health promotion programmes, including mental illness prevention interventions, help increase levels of mental health literacy in community members. While there is evidence showing the effectiveness of mental health promotion, much of what is known about this field is informed by studies conducted in high-income countries. There is a need to gather evidence about the effectiveness of such interventions in low- and middle-income countries (LMICs) where mental health services are often inadequate. In this systematic review, we synthesised the available published primary evidence from sub-Saharan Africa (SSA) on the types and effectiveness of mental health promotion programmes for young people. We performed a search of selected global databases (PubMed, PsycINFO, ScienceDirect and Google Scholar) and regional databases (Sabinet African Journals). We included observational, mixed methods, trials, pilots and quantitative original papers published from 2013 to 2023. We used the Mixed Methods Appraisal Tool (MMAT) to evaluate the quality of methods in selected studies, and the Preferred Reporting Items for Systematic Reviews and Meta-Analyses statement (PRISMA-2020) for reporting the evidence gathered. We identified 15 types of youth mental health promotion and illness prevention interventions. Among those identified, we found that school-based interventions enhanced mental health literacy, mental health-seeking behaviours and self-assurance and confidence among young people. Family-based interventions also showed a potential to improve relationships between young people and their caregivers. Future studies should explore how to further strengthen school- and family-based interventions that promote mental health among young people.

## Impact statement

Recent studies show high levels of mental health problems among young people in sub-Saharan Africa (SSA). Despite the high prevalence of mental health problems and the resultant consequences for young people, the provision of mental health services in the region remains poor. Mental health promotion is an effective intervention that can help prevent the onset of serious mental problems. This systematic review synthesised the available published primary evidence from SSA on the types and effectiveness of mental health promotion programmes for young people. Our review shows that school-based interventions increased mental health literacy among young people. In addition, young people who took part in school-based intervention programmes tended to be more self-assured and confident. Our findings also point to the importance of family-based interventions as these have the potential to improve relationships between young people and their caregivers. This review highlights the need for more evidence on the effectiveness of school- and family-based intervention programmes for young people in SSA.

## Background

Mental health refers to the state of well-being of individuals and encompasses an individual's ability to cope with the diverse stressors the individual faces (Herrman and Jané-Llopis, 2012). In defining mental health, Herrman and Jané-Llopis (2012) further added the concept of mental health promotion, which is a global initiative to improve and sustain mental well-being across different populations. The promotion of mental health incorporates the prevention of mental illnesses before their onset. Herrman and Jané-Llopis (2012) hold the view that mental health promotion requires an inclusive knowledge of determinants of mental health and mental problems with the sole purpose of preventing mental illnesses or promoting mental well-being for individuals.

According to a report by the World Health Organization (WHO, 2022), mental health problems have been on the increase largely due to the COVID-19 pandemic, which has created a crisis for mental health globally. This report further estimates that there was a sharp rise in anxiety and depression by more than 25% during the first year of the pandemic. Earlier in 2019, the Global Burden of Diseases, Injuries, and Risk Factors Study (GBD) showed that mental health problems remained among the top 10 leading contributors to the burden of disease globally, with anxiety and depressive disorders emerging as some of the most prevalent conditions (GBD 2019 Mental Disorders Collaborators, 2022). A systematic review and meta-analysis conducted during the COVID-19 pandemic estimated the global prevalence of mental health problems as follows: depression (28.0%), anxiety (26.9%), post-traumatic stress symptoms (24.1%), stress (36.5%), psychological distress (50.0%) and sleep problems (27.6%; Nochaiwong et al., 2021). Although mental health problems are prevalent globally for the general population, the situation is especially concerning for children and young people who are more vulnerable to developing these conditions (Mabrouk et al., 2022; Patel et al., 2008a; Patel et al., 2008b).

In sub-Saharan Africa (SSA) where more than 70% of the population is comprised of children and young people (Awad, 2019), the outlook is even more dire. For instance, a recent systematic review by Jörns-Presentati et al. (2021) found significantly high levels of mental health problems among adolescents with anxiety disorders estimated at 40.8% followed by depression at 29.8%, and emotional and behavioural problems at 21.5%. This high level of mental health problems among young people in SSA is further exacerbated by numerous psychosocial stressors, such as chronic poverty, prolonged exposure to war and violence and the high prevalence of HIV/AIDS in the region (Jörns-Presentati et al., 2021).

Despite the higher prevalence of mental health problems and the resultant consequences for young people, their families and the community, mental health services in SSA remain poor (Patel et al., 2008a; Patel et al., 2008b; WHO, 2020). Many countries in this region have poor mental health infrastructure (WHO, 2020), with deficient or non-existent mental health policies to address the mental health challenges faced by the communities (Sodi et al., 2021). The scarcity of mental health services (WHO, 2020) and the relatively high burden of disease (Jörns-Presentati et al., 2021) call for the implementation of innovative, evidence-based and culturally relevant interventions to promote mental health among young people in the region.

Growing evidence shows that mental health promotion, which includes mental illness prevention interventions, is an effective strategy that can reduce the burden of mental disorders and improve overall well-being in both children and adults (Barry et al., 2013; Castillo et al., 2020; Mabrouk et al., 2022; Singh et al., 2022; Teixeira et al., 2022). Mental health promotion is an area of public health practice that seeks to empower people to achieve positive mental health by encouraging healthy behaviours and addressing the needs of those susceptible to experiencing mental health problems (Barry et al., 2013). Based on the same concept of health promotion as articulated in the Ottawa Charter, mental health promotion advocates a population-based approach that seeks to build capacity in individuals and communities for well-being instead of focusing on ill health and the associated risks (WHO, 1986). In other words, such interventions tend to shift the focus from an individual to the broader community and the wider social determinants of mental health. Mental health promotion interventions, therefore, encourage broad public participation since they can be delivered in different settings, such as schools, the workplace and recreational centres (WHO, 2009). For instance, Santre (2022) pointed out that school-based mental health promotion programmes, such as social and emotional learning (SEL), mindfulness and positive psychology interventions, improve mental health, well-being and educational outcomes. A recent systematic review that sought to gather evidence on the cost-effectiveness of mental health promotion and prevention found that these interventions demonstrate good value for money when targeting children, adolescents and adults (i.e. Le et al., 2021). Apart from promoting high levels of mental well-being and preventing the onset of mental health conditions, mental health promotion also helps to increase levels of mental health literacy in society (Curran et al., 2023; WHO, 2009; Zhang et al., 2021, 2023).

"Although there is evidence showing the effectiveness of mental health promotion, much of what is known about this field is informed by studies conducted in high-income countries" (Erskine et al., 2017). A systematic review conducted more than a decade ago by Barry et al. (2013) synthesised findings on the effectiveness of mental health promotion interventions for young people (aged 6–18 years) in school- and community-based settings in low- and middle-income countries (LMICs). Given the availability of more recent data, there is a need for current evidence about the effectiveness of such interventions in LMICs, particularly the sub-Saharan region where mental health services are often inadequate (WHO, 2020). In SSA, such efforts should prioritise young people who constitute a great majority of the population (Awad, 2019).

## Aim

The aim of this study was to synthesise the available published primary evidence from SSA on the types and effectiveness of mental health promotion programmes for young people.

### Main questions

1. What types of mental health promotion programmes for young people have been implemented in SSA?
2. How effective are mental health promotion programmes for young people in SSA?

## Methods

We used a systematic review due to its ability to synthesise studies that have been done on any topic in a more detailed, meticulous and rigorous research methodology (Caldwell and Bennett, 2020). The review was guided by the Preferred Reporting Items for Systematic

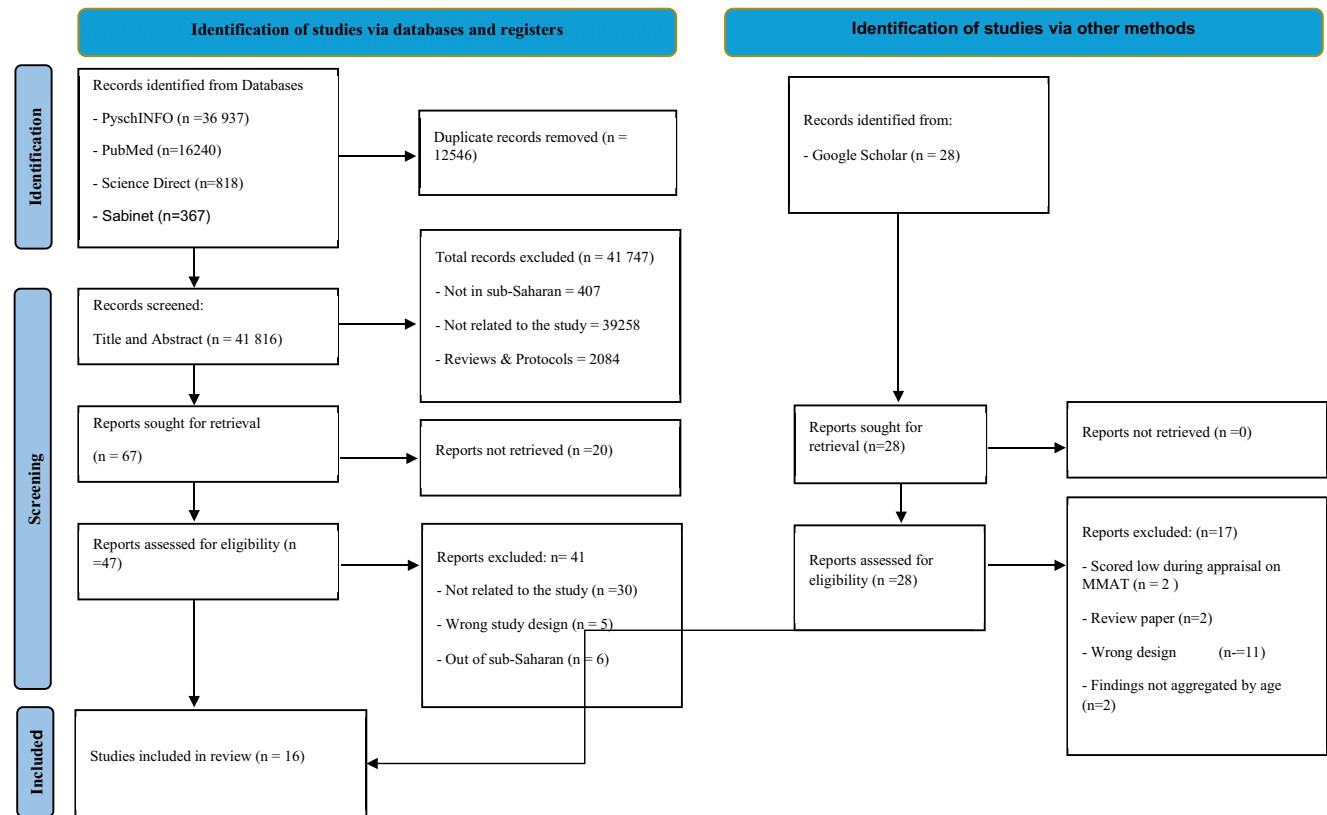

**Figure 1.** PRISMA diagram flow.
*Source*: Page et al. (2021)

Reviews and Meta-Analyses (PRISMA) guidelines (Page et al., 2021; see Figure 1). Global databases (PubMed, ScienceDirect and PsycINFO) and a regional database (Sabinet African Journals) were searched for data that were published between 2013 and 2023 without any design restrictions. This period was decided for the review in order to get the latest data on types of mental health promotion programmes for young people in SSA, including the effectiveness of these interventions. In this review, we adopted the WHO's definition of young people as individuals aged 10–24 years (WHO, 2023). We performed a reference and hand search on Google Scholar. We used search items: ("mental health" OR "mental disorder" OR "mental illness") AND (prevention OR "health promotion") AND (Type OR typologies) AND (effective*) AND ("Central Africa" OR "Africa South of the Sahara" OR "West Africa" OR "Western Africa" OR "East of Africa" OR "Eastern Africa" OR "Southern Africa" OR "sub-Saharan Africa"). We included studies conducted in any of the countries within the SSA region, focusing on young people regardless of gender, religion or sexual orientation. We included clinical and non-clinical studies involving mental health promotion programmes. We considered primary studies that employed quantitative, trials, pilot, mixed methods research approaches and observational studies.

We relied on the reference manager, EndNote20, to record all the identified articles on databases. Authors, KR, FKM, DI, PW, SOD, UI, DF, PH, MB and PT were all involved in the process of screening articles for eligibility. When there was conflict, TS was responsible for resolving the conflict and finding consensus. Articles that met the inclusion criteria were appraised by FKM and KR using the Mixed Methods Appraisal Tool (MMAT). The articles that met the inclusion criteria are listed on the data chart

**Table 1.** Heterogeneity statistics

| Tau | Tau² | $I^2$ | $H^2$ | $R^2$ | df | Q | p |
|-----|------|-------|-------|-------|-----|---|---|
| 0.205 | 0.0419 (SE= 0.0151) | 99.26% | 134.566 | . | 16.000 | 3414.711 | <0.001 |

(Annexure A, Table 3). We relied on the software Jamovi to determine the heterogeneity statistics of the selected studies (see Table 1). The p-value (<0.001), $I^2$ (99.26%) and $H^2$ (134.566) have higher values which indicate that there is substantial heterogeneity of the analysed data. We then applied narrative synthesis to discuss the types and effectiveness of mental health programmes that promote youth mental health in SSA.

## Results

### Characteristics of included studies

The attached Annexure B showcases the search history of the search. The search was conducted in 2023 and about 54,390 articles were found using PubMed, ScienceDirect, Sabinet African Journal, PSYCHINFO and Google Scholar (used for hand search, backward and forward search). Articles were then exported to EndNote, wherein duplicates were removed, and 41,816 articles were screened using title and abstract. Of the 41,816 articles which were screened, 47 articles were selected for full-text screening. In the end, a total of 16 studies conducted in 18 sub-Saharan countries were included in the final review (Figure 1). Detailed characteristics of the included studies are presented in the data chart in Table 3 of Annexure A.

Studies included in our review were drawn from Nigeria ($n = 3$), Rwanda ($n = 1$), Burundi ($n = 1$), Tanzania ($n = 2$), South Africa ($n = 3$), Kenya ($n = 2$), Burkina Faso ($n = 1$), Uganda ($n = 1$), Botswana ($n = 1$) and Malawi ($n = 1$). One of the 16 studies was a multi-country investigation that included Tanzania and Malawi. In terms of the study design, most of the included studies were either randomised control trials ($n = 10$) or quasi-experimental designs ($n = 3$). Other study designs included pre–post experimental design ($n = 1$), and mixed methods design ($n = 2$).

### Types of intervention from included studies

In the analysed articles, we found that 16 different interventions categorised as family-orientated ($n = 3$), school-orientated ($n = 8$), peer-orientated ($n = 4$) and online-orientated interventions ($n = 1$) were used among youth in the SSA region. The family-orientated interventions included family strengthening intervention (FSI), VUKA family program and READY. Interventions that targeted learners in school were class-based intervention (CBI), school-based intervention, school-based educational intervention, school support intervention, school-based training programme on depression, living well post-intervention (life-skills intervention) and mental health teaching programme. Interventions that were peer-orientated were Sauti ya Vijana (SYV) intervention, Balekane EARTH programme, group-based intervention and intervention targeting grief and depression. Finally, for online-oriented methods, we found that researchers used mobile phone-based mental health interventions.

### The effectiveness of interventions

#### Effects on attitude, knowledge and behaviour

The study conducted by Atilola et al. (2022) in Nigeria reported that effect size ($\eta^2$) was highest for knowledge (students: 0.07, $p = 0.001$; teachers: 0.08, $p < 0.000$) and least for attitude (students: 0.003, $p = 0.002$ teachers: 0.085, $p = 0.06$). In addition, Kutcher et al. (2015) conducted a study in Malawi and Tanzania and reported an effect size of $d = 1.16$ on knowledge which indicated that the training had a substantial impact on educators' knowledge acquisition. Also, it was reported that the trial had a positive impact on attitudes towards mental health with an effect size of ($d = 0.79$). This demonstrates a large increase in educators' positive attitudes and a decrease in stigmatising attitudes. Furthermore, Oduguwa et al. (2017) reported that attitude scores in the intervention group have an increase from 4.9 at baseline to 5.8 post-intervention ($p = 0.004$). Finally, McMullen and McMullen (2018) reported on prosocial attitudes/behaviours with a small effect size of, $F(1,167) = 5.61$, $p = 0.019$, $\eta^2 = 0.033$, and connectedness with a large effect of, $F(1,167) = 15.24$, $p < 0.001$, $\eta^2 = 0.085$. Bhana et al (2014) did not have numeric numbers to report intervention effect size but also noted a general improvement in mental health and cited examples of improvement in attitude towards HIV treatment knowledge. In terms of behaviour, Thurman et al.'s (2017) study reported that behaviour after intervention in adolescents was lower in the intervention group ($p = 0.017$, $d = –0.31$; Thurman et al., 2017).

#### Effects on perseverance, self-esteem and confidence

When assessing the intervention effect size, Betancourt et al. (2014) reported a significant improvement in children's perseverance and self-esteem (6-month follow-up: $d = 0.853$, $d = 0.853$). Ismayilova et al. (2018) also reported improvement in self-esteem at 12 months: small effect size Cohen's $d = 0.21$ and improvement in self-esteem at 24 months: Cohen's $d = 0.21$. In support, McMullen and McMullen (2018) reported medium effect sizes for general self-efficacy, $F(1,167) = 19.66$, $p < 0.001$, $\eta^2 = 0.106$, and internalising problems, $F(1,167) = 10.58$, $p = 0.001$, $\eta^2 = 0.060$. Finally, Kachingwe et al.'s (2021) study established confidence by reporting the effect size of 0.098 on confidence and psycho-social well-being.

#### Effects on depressive, anxiety and trauma

Betancourt et al. (2014) reported reductions in symptoms of depression (6-month follow-up: $d = -0.618$, $d = -0.618$), anxiety/depression (6-month follow-up: $d = -0.640$, $d = -0.640$) and irritability (6-month follow-up: $d = -0.788$, $d = -0.788$). On the other hand, Ismayilova et al. (2018) in a study conducted in Burkina Faso reported a reduction in depressive symptoms at 12 months: medium effect size Cohen's $d = -0.41$, reduction in depressive symptoms at 24 months: Cohen's $d = -0.39$. Thurman et al.'s (2017) study had an effective size of ($p = 0.009$, $d = –0.21$) on depression. Again, Green et al (2019) reported a moderate effect size of the intervention on depression among adolescent orphans in the study with $-0.28$, with a 95% confidence interval ranging from $-0.45$ to $-0.12$. In terms of trauma symptoms, the intervention effect size reported by Ismayilova et al. (2018)'s study indicated that there was a reduction in symptoms of trauma at 12 months: incidence risk ratio (IRR) = 0.62.

#### Effects on grief and *resilience*

The waitlisted group of participants reported an effect size for intrusive grief ($p = 0.000$, Cohen's $d = –0.21$) and complicated grief ($p = 0.015$, $d = –0.14$; Thurman et al., 2017). Katisi et al.'s (2019) study also reported small-to-moderate improvements in resilience and grieving among participants. For example, in the case of resilience, the effect sizes ranged from $r = 0.10$ to $r = 0.14$. In terms of grieving, effect sizes were reported as $r = 0.03$ to $r = 0.41$. The study reported that the overall resilience of the participants showed that males had slight improvement.

#### Effects on family *communication* outcome

Puffer et al (2016) reported positive outcomes on family communication, high self-efficacy for risk reduction skills and HIV-related knowledge and reduced high-risk behaviours.

### Publication bias assessment

We applied a fail–safe N analysis to check for publication bias. The result in Table 2 shows that the fail–safe N is 129,365.000 with the $p$ value at <0.001, which indicates that publication bias was avoided and suggests that the effectiveness of the analysed studies was robust and not dependent on the number of studies included in the analysis (see also funnel plot in Figure 2).

Fail–safe analysis was done using the Rosenthal Approach ($p < 0.001$). From the funnel plot, there is an apparent symmetry which shows that publication biases have been avoided.

**Table 2.** Fail–safe analysis

| Fail–safe *N* analysis (file drawer analysis) | |
| --- | --- |
| Fail–safe *N* | *p* |
| 129,365.000 | <0.001 |

*Note*: Fail–safe *N* calculation using the Rosenthal approach.

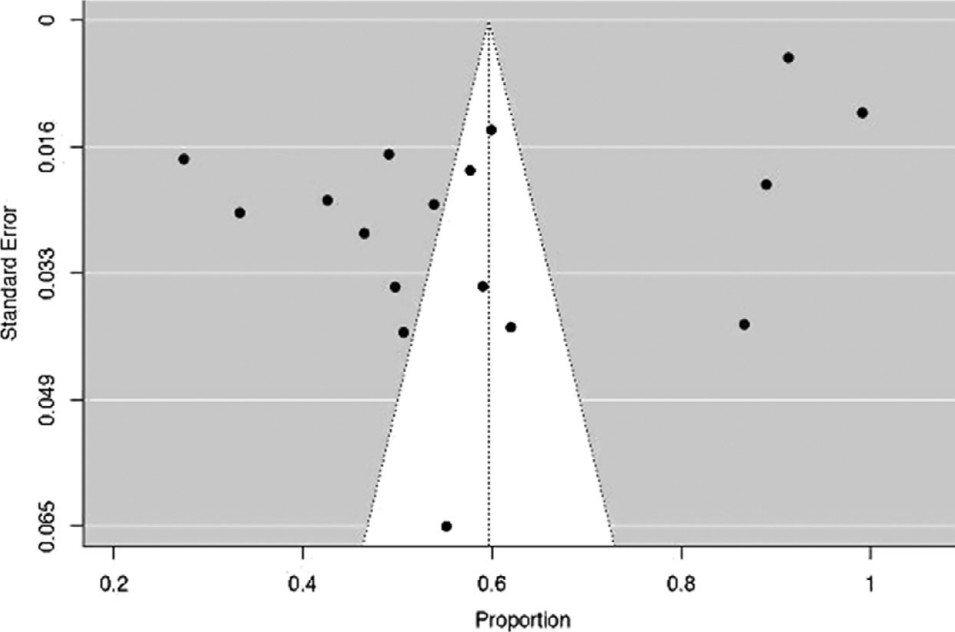

**Figure 2.** Funnel Plot.

## Discussion

In this review, we sought to determine the types and effectiveness of mental health programmes that promote mental health among young people in SSA. Such initiatives are a proven effective strategy to reduce the burden of mental illness among young people in the region, especially as studies have shown that timely interventions during this developmental period can help to reduce the risk of mental ill-health (Colizzi et al., 2020; McGorry and Mei, 2018; Saxena et al., 2013) and increase the prospects of a healthy adulthood. To our knowledge, this is the first such systematic review on the effectiveness of mental health promotion programmes among young people specifically in SSA despite there being several such programmes in this region (Atilola et al., 2022; Bella-Awusah et al., 2014; Thurman et al., 2017). This study thus builds on a previous systematic review by Barry et al. (2013) that explored the effectiveness of youth mental health promotion interventions in LMICs. In that review, Barry et al. (2013) established that school-based interventions have a positive impact on the mental health of adolescents as it improves their self-esteem. Betancourt et al. (2014) further found that as with school-based intervention, FSI and trickle-up intervention (Ismayilova et al., 2018) had a positive impact on the self-esteem of adolescents.

In addition, we found that school-based mental health interventions such as those conducted by Bella-Awusah et al. (2014), Thurman et al. (2017) and Atilola et al. (2022) improved the mental health literacy of young people. However, hitherto, no subsequent studies in SSA have explored the impact of the whole school approach interventions (Barry et al., 2013). This consequently leaves a gap as the whole-school approach has been reported to have a long-term impact on the mental well-being of adolescents than single-school interventions.

The findings by Bella-Awusah et al. (2014), Thurman et al. (2017) and Atilola et al. (2022) are consistent with those of studies conducted by Amado-Rodríguez et al. (2022) and Curran et al. (2023), which reported that mental health literacy interventions are effective in augmenting mental health knowledge and reducing

stigma. This finding offers an important avenue to support the mental health of young people since previous studies have shown that young people in SSA have low levels of mental health literacy (Wadende and Sodi, 2023). Young people with enhanced mental health literacy easily seek and effectively utilise professional mental health care for themselves and others (Colizzi et al., 2020; McGorry and Mei, 2018; Saxena et al., 2013). Being able to actively seek mental health care is especially important when young people live in LMIC contexts characterised by destabilising forces such as conflict and poverty and related deprivation that easily predispose them to mental illness (Kieling et al., 2011). Further, the limited capacity of such contexts (paucity of mental health care workers) and the stigma associated with the illness (Osborn et al., 2021) underscore the importance of a young population that has high levels of mental health literacy.

In addition to the mental health interventions increasing related literacy among participating young people (Amado-Rodríguez et al., 2022; Curran et al., 2023; Zhang et al., 2021; 2023), those who took part in intervention programmes exhibited increased self-assurance and confidence, which consequently had a positive impact on their mental health. It is therefore important to foster more of such interventions for the young people to ensure the future of SSA, where future leaders and the general society have the desirable emotional fortitude needed in their daily encounters.

Another finding from the study is that family-based intervention programmes had the capacity to strengthen family relations by improving the bond between young people and their caregivers. Unresolved mental health problems in young people continue into their adulthood negatively impacting their relationships, productivity and overall quality of life (Colizzi et al., 2020; McGorry and Mei, 2018; Saxena et al., 2013). The importance of family and related care relationships for the continued health of young people as they grow into adulthood cannot be overstated. Several studies indicate that there is a positive link between supportive family relationships and mental well-being (Daines et al., 2022; Chen and Harris, 2019). Such positive supportive relationships are also

essential especially when a young person is undergoing a mental health intervention program. They help the young person not to relapse during or after the programme ends (Puffer et al., 2016). Therefore, there is a need to invest more in such programmes that bring young people and their families much closer to improving their mental well-being.

Although it is not possible to draw any conclusions based on one study, Mindu et al. (2023) suggested that mobile phone-based interventions could be another resource that can be used in mental health promotion programmes for young people. While mobile phone-based interventions may have the potential, it is also important to take into account the limitations of such an intervention due to challenges such as language barriers, limited privacy and such interventions being perceived as not user-friendly (Mindu et al., 2023). Mobile phone coverage also remains a challenge for large swathes of populations in Africa. As estimated by GMSA state of mobile connectivity 2022, Africa has a 17% gap in coverage, and for the remaining 83%, there is a 61% usage gap where hundreds of millions are covered but not using the mobile internet (Gilbert, 2022).

## Limitations, future directions and conclusion

This systematic review identified the types of mental health intervention programmes for young people in SSA, assessed their effectiveness and identified gaps in the existing literature while highlighting areas that may need further research. The strengths of this review include the wide range of databases we searched. We obtained evidence from international databases (PubMed, ScienceDirect and PsycINFO) and regional African databases (Sabinet African Journals). Additionally, we did not set any restrictions on study design which is also a strength of our review. However, a limitation of this review is that it comprised only manuscripts published between 2013 and 2023; hence, the findings may not be applicable to any other period. We also did not include manuscripts published in other languages apart from English, which could likely result in some relevant evidence being omitted by our review.

Future studies could explore how to further strengthen school-based interventions, particularly whole-school approaches for promoting mental wellbeing and illness awareness among young people. This is of importance as our review shows its potential to have a sustainable impact on the mental health of young people, particularly when their mental health literacy is improved. Additionally, family-based interventions could be further developed and employed as our review revealed their potential to improve relationships between young people and their caregivers, thus promoting healthier families and, subsequently, whole communities. In SSA, mobile phone platforms have the potential to be useful and cost-effective avenues for mental health interventions targeting young people due to the wide use of mobile telephones even in the remotest locations. Researchers should look for creative ways to minimise the perceived impediments to the use of mobile phone platforms among young people.

**Open peer review.** To view the open peer review materials for this article, please visit http://doi.org/10.1017/gmh.2024.153.

**Data availability statement.** All data relevant to this review are available within the published manuscript and its online supplements.

**Author contribution.** TS conceptualised the idea, wrote the original draft, reviewed all drafts, supervised the development of systematic review protocol, developed a search strategy for systematic review and served as guarantor for the contents of this paper. KR contributed to writing the original draft, developed search strategy for systematic review, contributed to writing original draft, performed preliminary literature search, conducted edits for the drafts, reviewed all drafts, reviewed and approved final draft. FKM wrote the original draft, developed search strategy for systematic review, performed preliminary literature search, conducted edits for the drafts, reviewed all drafts, reviewed and approved the final draft. PW contributed to writing the original draft, performed preliminary literature search, reviewed all drafts, reviewed and approved the final draft. DI contributed to writing the original draft, performed preliminary literature search, conducted edits for the drafts, reviewed and approved the final draft. SOD performed preliminary literature search, conducted edits for the drafts, reviewed and approved the final draft. DF performed preliminary literature search, conducted edits for the drafts, reviewed and approved the final draft. PH performed preliminary literature search, conducted edits for the drafts, reviewed and approved the final draft. UI performed preliminary literature search, conducted edits for the drafts, reviewed and approved the final draft. MB performed preliminary literature search, conducted edits for the drafts, reviewed and approved the final draft. DM conducted edits for the drafts, reviewed and approved the final draft. SE performed preliminary literature search, reviewed and approved the final draft. TP performed preliminary literature search, reviewed and approved the final draft. DJ reviewed and approved the final draft. JYP reviewed and approved the final draft. TA reviewed and approved the final draft. ERB reviewed and approved the final draft. LG conducted edits for the drafts, reviewed and approved the final draft.

**Financial support.** This research did not receive any specific grant from funding agencies. T.S. received funding from the South African Medical Research Council (SAMRC) and National Research Foundation of South Africa (NRF) Grant number 150571.

**Competing interest.** None.

**Ethics statement.** We did not seek ethical approval from any ethics committee as this is a systematic review of available and accessible literature. We registered the protocol with PROSPERO (Registration number: CRD42023434887).

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

## Annexure A: Data extraction

**Table 3.** Data chart

| Author(s) (year), country | Title of the study | Target population | Study design | Instrument used | Type/name of intervention | Frequency, time between baseline and follow up | Outcome | Effect size | Quality assessment |
|---|---|---|---|---|---|---|---|---|---|
| Bella-Awusah, et al. (2014), Nigeria. | The impact of a mental health teaching programme on rural and urban secondary school students' perceptions of mental illness in southwest Nigeria | 154 secondary school students Age ranging from 10 to 18 years | A quasi-experimental two group pre-test–post-test control group design (not blinded) | UK Pinfold questionnaire | Mental health teaching programme *Intervention name: not provided* | 6 months Students completed pre-, immediate post and 6 months post-intervention | • Significant difference between intervention and control groups with intervention group's mean knowledge scores showing improvement<br>• No significant difference in attitude and social distance scores between the two groups<br>• There was a change in the intervention group wherein participants gained knowledge that mental health is caused by stress and that there is stigma on people with mental health<br>• There were no changes in attitude that mental illness is caused by spirituality and heredity<br>• There was a positive change in their perception on the causes of mental illness but no changes were recorded on the attitude and social distance towards mental illness<br>• There was an improvement in the perception that mental-ill persons are difficult to talk to | No effect size was reported | 6/7 |
| Betancourt et al. (2014), Rwanda | Family-based prevention of mental health problems in children affected by HIV and AIDS: an open trial | 20 families (35 young and 26 caregivers) | Pre–post experimental design | Semi-structured interview (recorded and translated to English) Comprehensive quantitative batteries (orally by bilingual – English/ Kinyarwanda) | Family-based intervention *Intervention name: family strengthening intervention (FSI)* | 6 months | • Families reported high satisfaction with the FSI, caregivers noted improvement in family connectedness, good parenting, perseverance or self-esteem and prosocial behaviour<br>• The treatment was seen as a success with the only concern being that some participants felt that there is a need for material support | There were significant improvements in children's perseverance and self-esteem (6-month follow-up; $d$ = 0.853, $d$ = 0.853) There were reductions in symptoms of depression (6-month follow-up; $d$ = −0.618, $d$ = −0.618), anxiety/ depression (6-month follow-up; $d$ = −0.640, $d$ = −0.640) and irritability (6-month follow-up: $d$ = −0.788, $d$ = −0.788) | 7/7 |

**Table 3.** (*Continued*)

| Author(s) (year), country | Title of the study | Target population | Study design | Instrument used | Type/name of intervention | Frequency, time between baseline and follow up | Outcome | Effect size | Quality assessment |
|---|---|---|---|---|---|---|---|---|---|
| Tol et al. (2014), Burundi | School-based mental health intervention for children in war-affected Burundi: A cluster randomized trial | 329 children | Cluster randomised trial (not blinded) | 14 focus group discussions Child Posttraumatic Symptom Scale | School-based intervention *Intervention name: class-based intervention (CBI)* | 5 weeks | • While there is no association between PTSD waitlist control conditions and living with both parents, depression symptoms tend to escalate over time for wait-list control condition participants who live with both parents<br>• During the treatment effect seen in longitudinal growth curve analysis, children in intervention group living in large families showed decrease in depressive symptoms<br>The effectiveness of school-based interventions depends on individual factors such as age, gender and family functioning | No specific effect size was mentioned on primary and secondary outcome measures This signified that intervention had no significant impact on the primary and secondary outcome | 7/7 |
| Kutcher et al. (2015), Malawi and Tanzania | Improving Malawian teachers' mental health knowledge and attitudes: An integrated school mental health literacy approach | 218 youth | Randomised control trial (not blinded) | Questionnaire with 7-point Likert Scale | School based educational intervention *Intervention name: African Guide: Malawi version (AGMv)* | Weekly | • Positive effect in improving mental health literacy among teachers and youth club leaders<br>• Prior to the training, the youth had a 58.3% knowledge about mental health. That increased a to 76.3% after training<br>• At baseline, the attitude towards MH was moderate but after training there was a significant increase in the attitude of youth towards MH<br>• The teachers developed a positive attitude towards MH and decrease in stigmatizing attitude towards MH after training | The study reported an effect size of $d = 1.16$ on knowledge which indicated that the training had a substantial impact on educators' knowledge acquisition. Also, it was reported that the trail had a positive impact on attitudes towards mental health with an effect size of $d = 0.79$. This demonstrates a large increase in educators' positive attitudes and a decrease in stigmatising attitudes | 7/7 |
| Bhana et al. (2014), South Africa | The VUKA family program: piloting a family-based psychosocial intervention to | 75 families | Pilot randomized control trials (not blinded) | Self-administered questionnaire | Family based intervention for early adolescents *Intervention* | 3 months | • Youth in the VUKA condition had greater improvements in reported ART adherence and treatment knowledge, | The numerical effect sizes were not specified in the study. However, the study reported general improvements in mental health, youth | 7/7 |

| Author(s) (year), country | Title of the study | Target population | Study design | Instrument used | Type/name of intervention | Frequency, time between baseline and follow up | Outcome | Effect size | Quality assessment |
|---|---|---|---|---|---|---|---|---|---|
| | promote health and mental health among HIV infected early adolescents in South Africa | | | | *name: VUKA family program* | | than the youth in comparison group • Caregivers in the VUKA intervention group reported significant change in comfort communicating with their children on sensitive topics | behaviour, HIV treatment knowledge, stigma, communication and adherence to medication | |
| Puffer et al. (2016), Kenya | A church-based intervention for families to promote mental health and prevent HIV among adolescents in rural Kenya: Results of a randomized trial | 440 (237 adolescent and 203 caregivers) | Cluster randomized trial design (not blinded) | Survey | A church-based intervention *Intervention name: READY* | 3 months | • Positive effect on Improving MH and preventing HIV infection among adolescents • READY increased family communication and protective factors against mental health disorder | • Ordinary least square was used to determine the intervention effect size • There were positive outcomes on family communication, high-self efficacy for risk reduction skills and HIV related knowledge and reduced high-risk behaviours | 7/7 |
| Thurman et al. (2017), South Africa | Effect of a bereavement support group on female adolescents' psychological health: A randomised controlled trial in South Africa | 453 bereaved students | Randomised controlled trial (not blinded) | Survey and face to face interviews | Intervention targeting grief and depression *Intervention name: Abangane* | 3 months | • Positive results from treatment group on grief and decreased behavioural problems reported • The intervention group showed significant decrease in depression and maladaptive grief • Caregivers who were part of intervention reported that there have been lower behavioural problems among the young people | • The waitlisted group of participants reported an effect size for intrusive grief ($p$ = 0.000, Cohen's $d$ = −0.21), complicated grief ($p$ = 0.015, $d$ = −0.14) and depression ($p$ = 0.009, $d$ = −0.21). Caregivers in the intervention group reported lower levels of behavioural problems among adolescents ($p$ = 0.017, $d$ = −0.31) | 7/7 |
| Oduguwa et al. (2017), Nigeria | Effect of a mental health training programme on Nigerian school pupils' perceptions of mental illness | 205 students | Randomised control trial (not blinded) | UK Pinfold questionnaire | School-based intervention *Intervention name: Not provided* | 6 months and 4 months | • Intervention group had a significantly higher mean knowledge score compared to the control group • About 20.8% of the 24 participants expressed that they were fearful after hearing the symptoms of mental illness • 92% of the youth indicated that the training programme was beneficial to them • Some pupils reported increased knowledge about mental illness during the discussion in the training | • The study revealed that the mean knowledge score in the intervention group increased significantly from 21.1 at baseline to 26.2 post-intervention ($p$ <0.001) as compared to the control group which has no significant change • Attitude scores in the intervention group has an increase from 4.9 at baseline to 5.8 post-intervention ($p$ = 0.004) | 7/7 |

| Author(s) (year), country | Title of the study | Target population | Study design | Instrument used | Type/name of intervention | Frequency, time between baseline and follow up | Outcome | Effect size | Quality assessment |
|---|---|---|---|---|---|---|---|---|---|
| Dow et al. (2018), Tanzania | Building resilience: A mental health intervention for Tanzanian youth living with HIV | 58 youth | Individually randomized group treatment | Pre-intervention questionnaire | Intervention for youth living with HIV *Intervention name:* (*Sauti ya Vijana:* The Voice of Youth) intervention | 6 months | • Successful according to its outcome measures, for example, feasibility demonstrated with good attendance of intervention sessions<br>• Acceptability was also high with youth enthusiastic about the programme<br>• Youth were very interested to learn about MH<br>• There was no difference in the type of stress experienced by both male and female participants<br>• Participants recommended new strategies of coping with stress such as breathing exercises<br>• Some of the male counterparts during group discussion indicated that they relied on alcohol in the morning to help reduce their worries<br>• The cognitive triangle suggested in the group discussion did not always work when dealing with bad thoughts<br>• After attending SYV, the participants went on to become peer youth leader<br>• Some of the caregivers indicated that they observed positive change in the attitude, confidence, and anxiety levels of the young people who attended the training<br>• With the aid of cognitive triangle, the participants were able to release their self-stigmatizing thoughts leading to change in their feelings and behaviours | The effective size was not mentioned | 5/7 |

*Cambridge Prisms: Global Mental Health*

| Author(s) (year), country | Title of the study | Target population | Study design | Instrument used | Type/name of intervention | Frequency, time between baseline and follow up | Outcome | Effect size | Quality assessment |
|---|---|---|---|---|---|---|---|---|---|
| Ismayilova, et al. (2018), Burkina Faso | Improving mental health among ultra-poor children: Two-year outcomes of a cluster-randomized trial in Burkina Faso | 360 children | Cluster-randomized trial (not blinded) | Interviewer-administered surveys | Group-based intervention (combination of economic strengthening interventions and family coaching) *Intervention name: Trickle Pu Plus/TU + arm)* | 12–24 months | • Children in Trickle Up (TU) intervention arm showed reduction in depressive symptoms compared to the control condition and economic intervention alone<br>• Due to the intervention, there was an improvement in self-esteem<br>• Self-esteem took longer for girls in the treatment as compared to girls in control group<br>• Boys reported higher self-esteem | • Reduction in depressive symptoms at 12 months: medium effect size Cohen' $d = -0.41$<br>• Reduction in depressive symptoms at 24 months: Cohen's $d = -0.39$<br>• Improvement in self-esteem at 12 months: small effect size Cohen's $d = 0.21$<br>• Improvement in self-esteem at 24 months: Cohen's $d = 0.21$<br>• Reduction in trauma symptoms at 12 months: incidence risk ratio (IRR) = 0.62 | 7/7 |
| McMullen and McMullen (2018), Uganda | Evaluation of a teacher-led, life-skills intervention for secondary school students in Uganda | 620 students | Cluster-controlled before-and-after study which used a wait-list control group (Not blinded) | Pre- and post-intervention questionnaires | Teacher-led life-skills intervention *Intervention name: Living Well intervention* | 6 months | • Intervention group showed a significant increase in general self-efficacy and significant reduction of internalizing problems<br>• There was an increase of self-efficacy, decrease in internalising depression- and anxiety-like symptoms<br>• There was a significant increase in connectedness in intervention as compared to control group | • The study reported effect sizes using Cohen's $d$ for different outcomes<br>• It reported medium effect sizes for general self-efficacy, $F(1,167) = 19.66$, $p < 0.001$, $\eta^2 = 0.106$, and internalizing problems, $F(1,167) = 10.58$, $p = 0.001$, $\eta^2 = 0.060$.<br>• Furthermore, prosocial attitudes/behaviour reported small effect size of, $F(1,167) = 5.61$, $p = 0.019$, $\eta^2 = 0.033$, and connectedness reported large effect, $F(1,167) = 15.24$, $p < 0.001$, $\eta^2 = 0.085$ | 7/7 |
| Katisi et al. (2019), Botswana | Fostering resilience in children who have been orphaned: Preliminary results from the Botswana Balekane EARTH program | 650 orphaned adolescents. | Quasi-experimental pre-test–post-test design (not blinded) | Self-report questionnaire | Resilience programme for orphaned children *Intervention name: Balekane EARTH program* | 2 weeks | • Improvements in reported resilience. Females reported improvements in problems related to grief and future aspirations<br>• Resilience was significantly improved<br>• Males and females did not report a significant reduction in grief at the end of the intervention | • The study reported small to moderate improvements in resilience and grieving among participants<br>• For example, in the case of resilience, the effect sizes ranged from $r = 0.10$ to $r = 0.14$<br>• In terms of grieving, effect sizes were reported as $r = 0.03$ to $r = 0.41$<br>• The study reported that overall resilience of the participants showed that males have slight improvement | 7/7 |

**Table 3.** (*Continued*)

| Author(s) (year), country | Title of the study | Target population | Study design | Instrument used | Type/name of intervention | Frequency, time between baseline and follow up | Outcome | Effect size | Quality assessment |
|---|---|---|---|---|---|---|---|---|---|
| Green, et al. (2019), Kenya | The impact of school support on depression among adolescent orphans: A cluster-randomized trial in Kenya | 835 participants | Cluster-randomized trial (not blinded) | | School support intervention *Intervention name: Not provided* | Once | • Intervention prevented depression severity scores from increasing over time among adolescents recruited from intervention schools<br>• There was no evidence of treatment heterogeneity by gender or baseline depression status | • The standardized effect size at the last wave (Year 4) was −0.28, with a 95% confidence interval ranging from −0.45 to −0.12<br>• This indicates a moderate effect size of the intervention on depression among adolescent orphans in the study | 7/7 |
| Kachingwe, et al. (2021), Malawi | Assessing the impact of an intervention project by the Young Women's Christian Association of Malawi on psychosocial well-being of adolescent mothers and their children in Malawi | 211 participants | Descriptive mixed methods evaluation study with intervention and control groups (not blinded) | baseline survey (administered using audio-assisted self-interview) and biomarker testing | Assessment of impact of a project on psychosocial well-being *Intervention name: Not provided* | 2 months | • The intervention group showed a statistically significant increase in knowledge of parenting skills, nutritional practice, motor skills and cognitive functions in children, as well as expressive language and socio-emotional capacities in children<br>• Psycho-social well-being was not statistically significant | • The study reported an effect size of 0.098 on confidence and psycho-social well-being<br>• The study reported an effect size of 21.91 on motor skills, cognitive functions | 6/7 |
| Atilola, et al. (2022), Nigeria | Towards school-based mental health programs in Nigeria: The immediate impact of a depression-literacy programme among school-going adolescents and their teachers | 3392 participants | Quasi-experimental post–pre interventional study Not blinded | Pre- and post-survey was administered to both student- and teacher-participants as part of the training | School-based training programme on depression-literacy *Intervention name: Break Free from Depression* | 3 months | • Participants reckoned that the intervention tool is ideal for learning about depression and sociality<br>• Students reported a positive change in the knowledge of, attitude towards, and confidence in handling depression<br>• Participants agreed to recommend the intervention programme to others since it worked for them | • The calculated effect size ($\eta^2$) was highest for knowledge (students: 0.07, $p = 0.001$; teachers: 0.08, $p < 0.000$) and least for attitude (students:0.003, $p = 0.002$ teachers: 0.085, $p = 0.06$) | 6/7 |
| Mindu, et al. (2023), South Africa | Digital mental health interventions for young people in rural South Africa: Prospects and challenges for implementation | 150 participants | Mixed-methods study (not blinded) | Questionnaire | Mobile phone-based mental health intervention *Intervention name: EMDIYA* | 1 month | Barriers impeding the uptake of digital health intervention include high cost of data, restrictive religious beliefs, limited privacy, lack of native languages on most digital platforms, low digital literacy and complicated user interface | The study did specify numerical effect size but reported that 50% of participants came across mental health apps but did not use them while 92% believed that digital apps could improve mental health literacy among youth | 6/7 |

## Annexure B: Search history

| Database | No of articles | Search history |
|---|---|---|
| Pubmed | 16242 | Search: (((((((((type) OR (classification)) AND (effective*)) AND ("mental health")) OR ("mental disorder")) OR ("mental illness")) AND (prevention)) OR (promotion)) AND (Africa)) OR ("sub-Saharan Africa") Filters: Humans, English, Female, Male, from 2013 – 2023 Sort by: Most Recent (((((((("type"[All Fields] OR ("classification"[MeSH Terms] OR "classification"[All Fields] OR "classifications"[All Fields] OR "classification"[MeSH Subheading] OR "classification s"[All Fields] OR "classificator"[All Fields] OR "classificators"[All Fields])) AND "effective*"[All Fields] AND "mental health"[All Fields]) OR "mental disorder"[All Fields] OR "mental illness"[All Fields]) AND ("prevent"[All Fields] OR "preventability"[All Fields] OR "preventable"[All Fields] OR "preventative"[All Fields] OR "preventatively"[All Fields] OR "preventatives"[All Fields] OR "prevented"[All Fields] OR "preventing"[All Fields] OR "prevention and control"[MeSH Subheading] OR ("prevention"[All Fields] AND "control"[All Fields]) OR "prevention and control"[All Fields] OR "prevention"[All Fields] OR "prevention s"[All Fields] OR "preventions"[All Fields] OR "preventive"[All Fields] OR "preventively"[All Fields] OR "preventives"[All Fields] OR "prevents"[All Fields])) OR ("promote"[All Fields] OR "promoted"[All Fields] OR "promotes"[All Fields] OR "promoting"[All Fields] OR "promotion"[All Fields] OR "promotional"[All Fields] OR "promotions"[All Fields] OR "promotive"[All Fields])) AND ("africa"[MeSH Terms] OR "africa"[All Fields] OR "africa s"[All Fields] OR "africas"[All Fields])) OR "sub-Saharan Africa"[All Fields]) AND ((humans[Filter]) AND (female[Filter] OR male[Filter]) AND (english[Filter]) AND (2013:2023[pdat])) |
| Science direct | 818 | **Search:** type OR classification) AND (effectiveness) AND ("mental health" OR "mental illness" OR "mental disorder") AND (prevention OR promotion) AND (Africa or "sub-Saharan Africa") |
| Sabinet | 367 | **Search:** [[All: type] OR [All: classification]] AND [All: effectiveness] AND [[All: mental health"] OR [All: "mental illness"] OR [All: "mental disorder"]] AND [All: prevention] OR [All: promotion]] AND [Africa] OR ["sub-Saharan Africa"]] (367) |
| PSYCHINFO | 36937 | Type OR classification AND effective* OR "mental health" OR "mental disorder" OR "mental illness" AND prevention OR promotion AND Africa OR "Sub-Saharan Africa" |
| Google scholar | 36 | Forward and backwards search |
| total | 10880 | |