## [Reviewer Report]

The authors are addressing an important topic, but I think the paper needs quite a bit more work before it is ready for publication.

Introduction needs to be clear that while this is apparently the first systematic review of studies to focus solely on SSA , it is not the first to focus on LMIC which of course includes SSA.

So last para of intro on page 4 should make it clear that there has already been one systematic review in LMIC , but not apparently since 2013.

Page 4, Main questions -it would be helpful to amplify question one of the study to “what settings, types and combinations of mhp….”, and to answer these aspects in the discussion.

Page 6…please explain internalising and externalising coping strategies, and also internalising disorders.

Need to adjust sentence on page 8 to acknowledge that the Barry et al 2013 study is in fact an earlier, indeed the first, systematic review of MHP in LMIC, which of course included SSA .

Not sure about the sentence on page 9 about future leaders….self assurance and confidence are of course needed for general societal reasons, not just for future leaders!

The discussion should make it clearer whether schools based programmes improve literacy, self assurance and confidence AND family relations , or whether it is only family interventions that improve family relations. ie should a country invest in both types of intervention or only in school interventions?

. The discussion should highlight the fact that the Barry review found that Whole School Approaches to mental health promotion were more effective than single school interventions . Did the present review contain any studies of the whole school approach or were they all single interventions, and therefore does it have any further evidence to bear on the issue of whether school based interventions should be WSA or more specific.

Middle para on page 9, please specify the settings in which these programmes happened.

Last para on page 9, first sentence “these intervention programmes”-specify which intervention programmes

Page 10 , end of first para, the last sentence needs to be rewritten to make grammatical sense…eg “This would allow” rather than “Thus allowing”

The middle para on page 10 should read “ hence the findings may NOT be applicable to any other period”

There should be more info in the table about the study design (eg selection of participants, assessment instruments, and what the intervention actually was ( eg rather than just class based intervention, or school based intervention). There is space for at least a couple of sentences into the relevant column here. Eg Green 2019 study in Kenya, we are given no information in the table on what is meant by school support.

Also need to know whether studies were double blinded or not.

It is relevant to have another column to know how long the time-period between baseline and follow up was, to know if changes are likely to be persistent. This is specified in the Ismayilove table entry but not in the others.

Table 20, the study is about teachers and students but the 3rd column only gives the number of students.

.

---

## [Reviewer Report]

This article reviews mental health promotion intervention literature from low- and middle-income countries. Results identified school- and family-based intervention approaches primarily, describing the studies and highlighting the limitations and opportunities in this area of work. This review fills a gap in the global mental health literature and brings attention to promotion interventions which are essential but often not prioritized.

Abstract: The abstract is clear and informative. The future directions sentence seems very specific only to the findings on school-based interventions. Authors should consider broadening this (at least to include family interventions, as it is in the impact statement) or clarifying that they are specifically recommending a specific focus on schools. Authors also write a generalized statement about phone interventions, though it looks like only one such intervention was included in the review.

Background: Authors provide helpful context in this section and clearly state the need for mental health promotion and for the review of the literature to date. One suggestion is to reduce the focus on the COVID-19 pandemic, however. While the pandemic certainly had an important and negative mental health impact, rates of mental health problems among children and adolescents were high before the pandemic. Authors should include statistics covering a longer period, especially because the literature included in this review dates back to 2013. It would be helpful for readers to understand the mental health needs present across the time covered by this review. As an additional minor suggestion, authors should consider beginning a new paragraph on line 55 where the focus shifts specifically to SSA.

Methods: Authors concisely report the key details of the methodology, with the use of PRISMA being a strength of the study. The number of search terms does seem relatively small, with not including a list of countries seeming potentially problematic if articles specifically mentioned a country setting without Africa as a prominent term (leading these not to be included). It would be helpful to know if authors involved a research librarian or other expert in systematic searches in developing this strategy. As a minor comment, some details on quality rating criteria included in the MMAT should also be included as all readers will not be familiar.

Results: The results section provides some helpful information but there are major concerns. Results lack detail that is needed to truly reflect the findings of this review, but there are also inconsistencies between the text and the table and inaccurate results included in the table. Steps for revision could include (1) first rechecking results of all articles and revising the table for accuracy, formatting, and consistency and (2) revising the text to accurately summarize the revised table content. Detailed comments here:

• Given that the results of the search before the 17 articles were chosen is important (e.g., how many articles were originally yielded by the search, most common reasons for exclusion), more text on this would be helpful even though it is largely covered in Figure 1. Figure 1 also seems very important to have in the main paper (noting since it is unclear if it is planned to be in the main paper or as an appendix).

• Major: Table 1 needs significant revisions for formatting and consistency in content (e.g., some quality rating cells have text while others have just the numbers; some results are exhaustive bullet points of all results while others are concise summaries).

• Major 1: Results of each study should be re-checked carefully. Some of the results are incorrect (e.g., Results of the READY trial did not show prevention of HIV infection). Some results are inconsistent with the table (e.g., text says that SYV reduced internalizing mental health disorders but the table does not show any findings related to that).

• Major: Authors should check the names of the interventions in text and table (e.g., Family Strengthen Intervention should be Family Strengthening Intervention; READY name should just be “READY” and does not include “family relationships” in the name). They should list the name before the acronym and then consistently refer to them the same way throughout (e.g., SYV by Dow et al. is one time referred to only as “Voice of Youth” when that is a secondary part of the title; the intervention by Puffer et al., READY, is one time referred to as “life skills intervention” which is not part of its name). Authors should also decide whether they are going to refer to them as the type of intervention versus the name and make sure that any interventions that are currently not named truly do not have a name (e.g., many of the school based interventions).

• Major: The section on effectiveness findings requires significant revision to reflect the results of the papers more accurately and comprehensively. This section is vague overall and does not speak to sizes of the effects. There are also sentences that seem more like discussion points than results (e.g., “With the increase in mental health literacy, young people are less likely to use internalising and externalising coping strategies when they are under emotional distress.”)

• It is unclear how Ayazi et al. (survey study) fits into the review given that it does not seem to be an intervention study.

• Minor: It looks as if the Funnel Plot may be planned for inclusion in the main paper, but it could be more helpful to put this in the appendix to make room for the Figure 1 and findings table to be included in the main paper. (It is currently unclear whether some of these elements will be supplementary material that some readers will not see.)

Discussion: The discussion is quite thin and repetitive with results, though specific suggestions are difficult to make before revisions to the results. Two initial comments: (1) Authors should connect for readers how improved family relationships indicate mental health promotion; there is literature to support this. And (2) As noted in the Abstract, authors should be cautious about making broader statements about phone-based interventions given the inclusion of only one study that used that modality.

---

## [Reviewer Report]

The authors have gone a long way to address my concerns. However, I still feel 3 points are not adequately addressed.

1. Since a key finding of the Barry et al review was not only that school based interventions have an impact, but also that the whole school approach has more of an impact than single school interventions, I think a key point to be made in the discussion of your review is that you found no subsequent SSA studies which explored this issue. It s a key message for future work.

2. the fact that only one study clearly set out the time between baseline and follow up measures is very concerning, and needs to be explicit in the paper.

3. I think the authors have misunderstood what I said about double blind RCTs...RCTs need to be double blind, so that neither the researchers nor the participants know whether they are in the intervention group or the control group. All RCT papers should specify this, so the review should comment if they have not. .

---

## [Reviewer Report]

The authors have been responsive to many of the comments by reviewers in relation to the text. However, major concerns still remain related to the Results. Some comments below represent more fundamental concerns while others relate to smaller details. However, in a systematic review, the details such as intervention names are quite important given that systematic reviews tend to be widely read and cited.

- Effect sizes are not described, which are such an important indicator of intervention effects.

- It seems that there may still be errors in the information presented in the “Outcome” column of the table based on a brief spot check. For example, in Oduguwa (2017), the Outcome in the table says “About 24 participants expressed that they were fearful after hearing the symptoms of mental illness.” In the original article, results state, “There were responses from 24 participants with 20.8% stating that hearing about the symptoms of mental illness had created fear in them.” It is possible that there is an explanation for this that the authors can clarify or address, but suggests that perhaps another check of all results of studies may need to be rechecked.

- Overall in the table, the Outcome column is inconsistent in content rather than presenting systematically similar information extracted for each study. This could be explained by sentences drawn from what the authors of the individual articles chose to emphasize in the text of their papers or abstracts, rather than authors determining which pieces of information to report across all. This is not typical of systematic review articles. Two examples include:

o Tol et al. (2014) intervention in Burundi: In outcome column, it says “The treatment was a success as there was an improvement in hope especially among younger children and those with low levels of exposure to traumatic events.” A review table typically should not label an intervention a success.

o Bhana et al. Outcome says “Positive results in both qualitative and quantitative measures on MH of adolescents with HIV” This relates to general mental health and is an incomplete sentence whereas, for other articles, outcomes are much more specific and phrased as different types of sentences.

- Minor: Program name consistency seems improved but may need to be checked again. I looked at just a few examples based on studies with which I am familiar and things that seemed simple to check. I found a few things that seem like errors that likely suggest another check needs to be done.

o I noticed Ismayilova’s study was puzzling since it was just named “group-based intervention” in the text. However, in the results table, the results description mentions Trickle Up.

---

## [Editor Report]

Unfortunately, we are unable to accept your manuscript in its current form. Kindly attend to the additional reviewer comments. In addition, please review p.10, lines 57-60, p.11, lines 4-5. “In that review, Barry et al. (2013) established that school-based interventions have a positive impact on the mental health of the adolescents’ as it improves their self-esteem. The findings were corroborated by Betancourt et al. (2014) who found that like school-based intervention, Family Strengthening Intervention, and group-based intervention (Ismayilova et al., 2018) had a positive impact on the self-esteem of adolescents”. This needs to be reworded to make better sense.

---

## [Reviewer Report]

This manuscript provides a valuable systematic review of mental health promotion interventions in Sub-Saharan Africa over the past decade. As promotion receives relatively little attention, this is an important contribution. However, the Results section of this manuscript is under-developed and difficult to follow.

The Introduction to this manuscript is clear and provides the appropriate information and rationale for the study.

Methods:

Many aspects of the methods are also clear. However, one missing piece is how authors defined “promotion” exactly, and how they defined “mental health.” This is very difficult to do within the field more broadly, especially when delineating promotion, prevention, and early intervention. Many interventions (including many reviewed here) are considered to fall into multiple of these categories. Likewise, there are other interventions I might have expected to see here that are not included (e.g., Shamiri in Kenya, Parenting for Lifelong Health in South Africa, Triple P applications in Africa – and these are just a few familiar to this reviewer) – but this is likely because of the way “promotion” and/or “mental health” were defined rather than any sort of oversight of the author team. It would be helpful to know if the search terms emphasized how the authors of the manuscripts named/labeled the intervention (specifically saying “promotion”), or on how they described the intervention, and/or on which outcomes they measured that implied an expected promotion outcome orientation.

Related, it would be helpful to clarify the search terms used across the databases versus those used only for google scholar, and to provide a brief rationale for adding google scholar in a different way. Authors should also mention how the screeners/coders were trained to be consistent in the decision rules they applied and data they extracted, as well as how they organized their review and extraction (e.g., with software designed for reviews, or perhaps excel databases that were shared or merged across those who participated).

Results:

The first two paragraphs of the Results and standard and clear. However, the entire section summarizing the effectiveness of intervention results is disjointed and difficult to follow. It is also all highlighted in yellow, raising the question of whether this was perhaps not the final draft of this section. The text provides statements about individual studies that seem accurate, and the organization of the paragraphs is logical. However, the paragraphs have no orienting or topic sentences and do not make any synthesizing statements related to the Results. Authors should revisit, rewrite, and expand this important section of the paper.

Discussion: The discussion points seem appropriate and provide some helpful synthesis of findings. It would be useful for authors to take the opportunity to add some further recommendations for future research based on the remaining gaps. A brief conclusion section would also be helpful.